# Blends of Human Milk Oligosaccharides Confer Intestinal Epithelial Barrier Protection In Vitro

**DOI:** 10.3390/nu12103047

**Published:** 2020-10-05

**Authors:** Jane M. Natividad, Andreas Rytz, Sonia Keddani, Gabriela Bergonzelli, Clara L. Garcia-Rodenas

**Affiliations:** 1Department of Gastrointestinal Health, Nestlé Research, 1000 Lausanne (Vaud), Switzerland; Sonia.keddani@rd.nestle.com (S.K.); Gabriela.bergonzelli@rdls.nestle.com (G.B.); 2Clinical Development Unit, Nestlé Research, 1000 Lausanne (Vaud), Switzerland; Andreas.rytz@rdls.nestle.com

**Keywords:** human milk oligosaccharides, Caco-2:HT29-MTX cultures, intestinal barrier function, permeability, inflammation

## Abstract

Breastfeeding is integral in the proper maturation of the intestinal barrier and protection against inflammatory diseases. When human milk (HM) is not available, supplementation with HM bioactives like Human Milk Oligosaccharides (HMOs) may help in providing breastfeeding barrier-protective benefits. An increasing HMO variety is becoming industrially available, enabling approaching the HMO complexity in HM. We aimed at assessing the impact of blends of available HMOs on epithelial barrier function in vitro. The capacity of individual [2′-Fucosyllactose (2′FL), Difucosyllactose, Lacto-N-tetraose, Lacto-N-neotetraose, 3′-Siallylactose and 6′-Siallylactose] or varying combinations of 3, 5 and 6 HMOs to modulate fluorescein-isothiocyanate (FITC)-labelled Dextran 4 KDa (FD4) translocation and/or transepithelial resistance (TEER) was characterized in Caco-2: HT29- methotrexate (MTX) cell line monolayers before and after an inflammatory challenge with TNF-α and IFN-γ. The six HMO blend (HMO6) dose-dependently limited the cytokine-induced FD4 translocation and TEER decrease and increased TEER values before challenge. Similarly, 3 and 5 HMO blends conferred a significant protection against the challenge, with 2′FL, one of the most abundant but most variable oligosaccharides in HM, being a key contributor. Overall, our results suggest differential ability of specific HMOs in modulating the intestinal barrier and support the potential of supplementation with combinations of available HMOs to promote gut health and protect against intestinal inflammatory disorders.

## 1. Introduction

Breastfeeding is the gold standard for infant nutrition [1]. The health relevance of human milk is corroborated by studies demonstrating that breastmilk feeding is associated with reduced risk to develop inflammatory diseases in infancy, such as infection, allergy and necrotizing enterocolitis, as well as later in life, such as rheumatoid arthritis and inflammatory bowel disease [2,3,4].

Several mechanisms, including maintenance of intestinal barrier integrity, are proposed to underlie the protective effect of human milk [5]. Shortly after birth, human infants have a permeable intestine, which becomes tighter within the first weeks of life [6,7]. Breast-fed infants have a faster evolution rate towards tighter epithelium than formula-fed infants [7,8]. It is not yet clear whether a delayed intestinal barrier maturation has an immediate consequence on infant gut health or may alter intestinal function programming that may, in turn, influence development and future health. Nevertheless, a dysfunctional epithelial barrier has been implicated as a critical factor in growth stunting [9] and in the predisposition to and pathogenesis of several intestinal and non-intestinal inflammatory diseases [10].

Although breastfeeding is integral for post-natal development and health later in life, many infants cannot be exclusively breast-fed due to variety of reasons and are instead exclusively or complementary fed with infant formula [11]. As such, an attempt to supplement infant formula with human milk bioactives has been increasingly encouraged in recent years. 

Human milk oligosaccharides (HMOs), a complex collection of structurally diverse, non-digestible carbohydrates, which collectively represents the third most abundant solid human milk element, are among the best-studied bioactives in human milk [12]. Increasing evidence suggests that multiple health benefits of breastfeeding may be related to HMO content and composition. For example, abundance of specific HMOs in maternal milk is inversely correlated to the risk of breast-fed infant infection [13]. HMOs are metabolized by the intestinal microbiota and their metabolites used as an energy source by both, the microbiota itself and the intestinal epithelial cells [14]. Intact HMO molecules can also survive the gastro-intestinal passage and microbiota fermentation [15,16], and thus may exert direct effects on the intestinal epithelium. A recent study reports that a complex HMO extract from pooled human milk samples increased intestinal mucin expression and decreased permeability in both a murine model of intestinal inflammation but also in vitro, in pathogen-challenged epithelial cell monolayers [17].

However, only select HMOs are currently available at the industrial scale. Several in vitro studies have reported the capacity of some of these HMOs, tested individually, to promote epithelial cell differentiation, maturation and mucin production, with effects that varied as a function of the HMO tested [17,18,19]. Specific HMOs have been also shown to differentially affect the transepithelial electrical resistance (TEER) in both pre- and post-confluent cells [18]. The functional relevance of these observations in terms of macromolecular permeability, particularly in the context of an inflammatory setting, is nevertheless unclear. Furthermore, it has not yet been addressed whether blends of industrially available HMOs, resembling more closely the human milk composition than individual HMOs, confer advantage in the modulation of the epithelial barrier integrity.

In this study, we characterized the impact on epithelial barrier function of HMOs industrially available and highly represented in human milk [Fucosylated: 2′-Fucosyllactose (2′FL); Difucosyllactose (DFL); Sialylated: 3′-Sialyllactose-(3′SL), 6′-Sialyllactose (6′SL); Non-fucosylated neutral: Lacto-N-neotetraose (LNnT), Lacto-N-tetraose (LNT)]. We hypothesized that each of these HMOs has a differential effect on epithelial permeability and that combinations of these molecules may, therefore, provide an optimal barrier protection against inflammation. Thus, the aim of this research was to determine whether a combination of these 6 HMOs (HMO6), in proportions resembling the natural occurrence in human milk [20], has the ability to reinforce the epithelium to protect the barrier integrity against a subsequent inflammatory challenge. In this study, we did not address the mechanism underlying the HMO6 effects, but instead we delineated whether specific HMOs predominantly drive the barrier effects by evaluating the impact of HMO6-derived sub-blends and individual HMOs.

## 2. Materials and Methods

### 2.1. Cell Lines

The human colorectal adenocarcinoma cell line Caco-2 (HTB-37) was obtained from the American Type Culture Collection (ATCC, Manassas, VA, USA) at passage 21 and used in experiments at passage 23 to 33. The human colon adenocarcinoma cell line HT29 (HTB-38; ATCC) previously adapted with methotrexate (MTX) was obtained from the European Collection of Authenticated Cell Cultures (ECACC, Salisbury, UK) at passage 51 and used in experiments between subsequent passages 23 and 33. Both cell lines were separately maintained in 75 cm^2^ tissue culture flasks (Fischer Scientific, Reinach, Switzerland) at 37 °C and 10% CO_2_, 95% air/water saturated atmosphere.

### 2.2. Cell Culture Model

Both Caco-2 and HT29-MTX cell lines were maintained in Dulbecco’s Minimal Essential Media (DMEM; 11965092, Gibco) supplemented with 10% (*v/v*) heat-inactivated fetal bovine serum (FBS; 10270-106, Gibco) and 1% (*v/v*) Penicillin and Streptomycin solution (P4333, Sigma-Aldrich, Saint-Louis, MO, USA). Growth medium was replaced at a minimum of twice per week. Cell lines were subcultured weekly at preconfluent densities with 0.4% trypsin–EDTA (T3924, Sigma -Aldrich, Saint-Louis, MO, USA).

For the experiments, Caco-2 and HT29-MTX cells were stained with trypan blue (T8154, Sigma-Aldrich, Saint-Louis, MO, USA), counted, resuspended in complete growth medium at ratios of 76:24, a ratio previously validated to assess transport and metabolism of food bioactives [21], and seeded at a density of 6 × 10^4^ cells per cm^2^ in Transwell™ Polycarbonate semi-permeable membrane of 0.4 µM pore size and 1.12 cm^2^ surface area (3460, Corning). Confluency and integrity of the Caco-2:HT29-MTX culture were evaluated by measuring manually the TEER every week using Millicell™ ERS-2 Voltohmmeter. Cells were used for experiments 21 days post seeding. Only cells with TEER between 600 and 800 Ω per cm^2^ were used in the experiments.

### 2.3. HMO Treatments and Inflammation-Induced Epithelial Barrier Dysfunction Model

Bacterially produced HMOs and, in one experiment chemically-synthesized 2′FL, were obtained from Glycom AS (Hørsholm, Denmark). HMOs as well as lactose (Sigma-Aldrich, Saint-Louis, MO, USA) were diluted in sterile water at a concentration of 250 mg/mL, sterile-filtered using 0.2 µM pore size and kept at −20 °C for a maximum of 6 months. On the day of the treatment, cell growth medium was replaced by fresh medium with all supplements but without phenol red. Blends of 2′FL, 3′SL, 6′SL, LNnT, LNT, DFL, the individual HMOs or lactose were further diluted in the growth medium and added in the apical compartment of the Transwell™ at varying final concentration (Table 1) before challenging with TNF-α and IFN-γ cytokines in the basolateral compartment. Prior to cytokine challenge, none of the individual HMOs or combination of HMOs significantly induced cell toxicity levels (less than 5% of lysed cells) as assessed by CytoTox 96^®^ Non-Radioactive Cytotoxicity Assay (Promega, Madison WI, US), even at the highest dosage, suggesting that the co-culture epithelial cell lines tolerate HMO exposure.

Controls were non-HMO treated cells and cytokine unchallenged cells (Control −) and non-HMO treated cells, but cytokine challenged (Control +).

A study was conducted to optimize the pro-inflammatory challenge settings, by evaluating the impact of the exposure time as well as of different TNF-α and IFN-γ concentrations and ratios on epithelial permeability. Combinations of TNF-α and IFN-γ at a concentration of 2.5 ng/mL and 10 ng/mL, respectively, for 48 h provided maximal differentiation in terms of permeability parameters between inflammatory challenged and non-challenged cells, with levels of cell cytotoxicity lower than 35% of lysed cells (data not shown). Similarly, a study conducted with HMO6 to determine the optimal HMOs’ treatment protocol indicated that 48 h exposure to HMO6 before and during the inflammatory challenge was able to modulate the permeability parameters (data not shown). As such, 48 h pre-treatment with HMOs followed by 48 h of TNF-α and IFN-γ challenge were used for the experiments in this study.

### 2.4. Epithelial Permeability Assessment

Permeability was assessed using two readouts: translocation of FITC-labeled dextran (FD4; 4000 Da, Sigma-Aldrich, Saint-Louis, MO, USA) from apical to basolateral compartment of the Transwell™ and, for specific experiments, TEER.

The FD4 assay was performed by adding in the apical compartment of the Transwell™ a solution of FD4 at a final concentration of 1 mg/mL. Basolateral samples (100 μl in duplicate) were collected before and 90 min after the apical addition of FD4 and fluorescence in the samples was measured (Em: 485 nm; Ex: 535 nm; Tecan Infinite 200). The FD4 concentration was calculated using a standard curve generated by serially diluting FD4 in culture media without phenol red and results were expressed as ng/mL. FD4 concentration was normalized relative to the value in the Control +. The Control + represents minimal resistance to FD4 while Control − represents maximal resistance to FD4 translocation.

TEER was dynamically measured every 5–15 min by placing the Transwell™ seeded with Caco-2:HT29-MTX culture in a cellZscope machine (Nano Analytics) inside at 37 °C and 10% CO_2_, 95% air/water saturated atmosphere for the whole duration of experiment. Growth medium and treatments were replenished every 48 h. TEER was measured as Ω per cm^2^ and percent change in TEER (% TEER change) was calculated relative to the baseline value (TEER measurement prior to any treatment).

### 2.5. Experimental Design and Data Analyses

The experimental design consists of 49 treatments varying systematically HMO composition and dosage, with comparisons to positive and negative controls. These 49 treatments are separated in 4 blocks (Table 1) with: (A) dose response for a mix featuring all six HMOs (HMO6) at ratios close to their relative concentrations in human milk [20]: 2′FL (55%), 3′SL (7%), 6′SL (9%), LNnT (5%), LNT (18%), DFL (6%); (B) all possible mixes of three HMOs (HMO3) at 60 mg/mL HMO6-equivalent dose (i.e., individual HMO ratios and concentrations similar to those found in HMO6 at dose 60 mg/mL), to estimate the individual impact of each HMO when present in combination with other HMOs; (C) all possible mixes of five HMOs (HMO5) at 30 mg/mL HMO6-equivalent dose (i.e., individual HMO ratios and concentrations similar to those found in HMO6 at dose 30 mg/mL) to estimate the effect of the absence of a single HMO in the mix; and (D) all single HMOs at both 30 mg/mL and 30 mg/mL HMO6-equivalent dose, to estimate the individual effect of each HMO, in comparison to lactose.

FD4 permeability was measured in duplicate or triplicate, and a minimum of two independent experiments were performed, except in the screening study in blocks B and C. These blocks were performed in a minimum of duplicate wells and conducted in a single experiment unless otherwise specified. In block A, TEER was assessed in a minimum of duplicate wells and in three independent experiments. In the other blocks, TEER was either not assessed or monitored in only one well in one experiment. These data are reported in Appendix A.

The analyzed endpoint across all blocks was FD4 translocation (relative to Control + = 100%, with standardization performed by plate) both before (for few selected treatments) and after challenge (all treatments). In addition, for block A, the mean TEER value before challenge (relative to baseline = 100%) and the mean TEER value after challenge (relative to baseline = 100%) were analyzed.

Differences between treatments were assessed using one-way analysis of variance and subsequent multiple comparisons using Fisher’s least significant difference procedure with α = 5% (LSD5%). This procedure uses the pooled standard deviation and the average number of replicates per treatment. Two treatments can be considered as statistically significantly different if the difference between their two means is larger than the LSD value. Mean values for all treatments are reported (Appendix A) together with pooled standard deviation, standard error and LSD5%. Mean values are also visualized (Figure 1, Figure 2, Figure 3 and Figure 4) using bar charts and error bars representing Mean ± 0.5 × LSD5%, allowing for a direct comparison of any pair of treatments (i.e., statistically significant difference if intervals do not overlap). Means of ternary mixes are not visualized as such, but as the estimated main effects of the individual HMOs in the mixes (i.e., mean of the group of 10 ternary blends featuring a given HMO).

All calculations were conducted with R 3.5.0 [22].

## 3. Results

### 3.1. A Blend of 6 HMOs Highly Represented in Human Milk Conferred Resistance against Inflammatory-Induced Epithelial Barrier Dysfunction

To determine the effects of a blend of 6 HMOs on epithelial permeability to macromolecules, the Caco-2:HT29-MTX cultures were treated with increasing concentrations of HMO6 (Table 1, Block A). Before the inflammatory challenge, pre-treatment with HMO6 did not significantly influence FD4 permeability compared to controls (Figure 1A, Appendix A). Upon cytokine challenge, Control + cells experienced a significant increase in FD4 translocation, with levels of cell cytotoxicity lower than 35% of lysed cells (data not shown). FD4 translocation was reduced by HMO6 in a dose-dependent manner, with a plateau reached after 30 mg/mL (Figure 1A, Appendix A). The TEER of the monolayers was further assessed at the most efficient HMO6 concentrations, to determine any impact of the treatment on ion permeability. Pretreatment with HMO6 increased the TEER values proportionally to its concentration (Figure 1B, Appendix A), and, upon cytokine treatment, limited the TEER decline and resulted in a significantly post-challenge mean TEER value higher than Control + at 60 mg/mL, the highest dose tested (Figure 1B, Appendix A).

The FD4 translocation test showing approximately 60% decrease of inflammation-mediated gut barrier disruption at a dose of 30 mg/mL was selected for the further experiments.

### 3.2. Combination of HMO Sub-Blends, Particularly Those Containing 2′FL, Protected the Epithelial Barrier

To identify the specific HMOs contributing to the observed effects of HMO6 and to explore any potential interaction between them, two complementary screening assays were performed on simpler, HMO6-derived sub-blends, by assessing the FD4 translocation after cytokine challenge.

In the first assay (Table 1, Block B), the mean FD4 translocation values of the 10 HMO3 blends featuring a given HMO were compared to those of the control monolayers by using a 60 mg/mL HMO6-equivalent dose. This dose was selected as leading to total HMO concentrations in all HMO3 groups close to 30 mg/mL, the minimal HMO6 dose conferring near maximal protection against FD4 translocation. Independently on their composition, FD4 translocation was lower in the cell monolayers treated with any of HMO3 blend groups than in Control + cells. When compared to the negative Control -, all the HMO3 blend groups displayed significantly higher permeability, except the group of 10 HMO3 blends containing 2′FL, which resulted in a maximal protection against the inflammatory challenge (Figure 2A, Appendix A).

In the second screening assay (Table 1, Block C), all the HMO5 sub-blends—lacking one specific HMO—conferred protection against cytokine-mediated permeability, as shown by the significant decrease of FD4 translocation values compared to Control +. Of note, only the one lacking 2′FL displayed significantly higher permeability values than HMO6 (Figure 2B, Appendix A).

Since the total HMO concentration of the HMO5 sub-blends was reduced proportionally to the concentration of the missing HMOs in the mixes, we sought next to confirm the observed effect of the HMO5 without 2′FL sub-blend by adjusting the concentrations of the 5 remaining HMOs to reach 30 mg/mL total HMO concentration (Table 1, Block C). Even in these conditions, removal of 2′FL from HMO6 again resulted in a significant reduction of the HMO6 protective effect against the inflammatory challenge (Figure 2B, Appendix A).

### 3.3. 2’FL Was the Most Efficient Single HMO in Preventing the Increased Epithelial Permeability after Inflammatory Challenge

To further confirm which HMOs were driving the blend barrier effects, cells were treated with individual HMOs and with lactose for comparison. HMOs were tested both (i) at concentrations mimicking their contribution to HMO6 and (ii) at iso-concentration settings (Table 1, Block D). 2′FL and, to a lesser extent, LNT, reduced FD4 translocation compared to Control + in a dose–response manner. No dose–effect was observed with LNnT, 3′SL, and 6′SL treatments, although they significantly reduced permeability at one dose or the two doses. By contrast, neither DFL nor lactose—the main carbohydrate of human milk—exerted any protective effect, even when tested at the highest dose of 30 mg/mL (Figure 3, Appendix A).

### 3.4. The Process of 2′FL Synthesis Does Not Influence Its Permeability Benefits

The HMOs tested in the experiments described above were derived from bacterial synthesis. To confirm that the results obtained were specifically due to the HMO molecule, and to rule out any potential impact of impurities generated during the fermentation process, we compared the barrier modulation capacity of two independent production batches of bacterial synthesized 2′FL to that of one batch of chemically synthesized 2′FL. Both, bacterial- and chemical-produced 2′FL displayed comparable ability to reduce FD4 translocation after the inflammatory challenge (Figure 4).

## 4. Discussion

Human milk has long been recognized as an early environmental factor that promotes a variety of health outcomes and reduces the risk of intestinal inflammatory diseases throughout life [23]. HMOs, a complex repertoire of chemically different carbohydrates, were identified as a contributing factor to the health outcomes attributed to human milk [13]. A few studies have reported positive effects of HMO extracts from human milk and of some single HMOs on intestinal epithelial cell functions and features including epithelial cell differentiation and maturation, as well as mucin production [17,18,19]. This substantiates the rational to supplement formula-fed infants with HMOs to be closer to the breastfeeding nutritional and health benefits. However, only select HMOs are currently available in volumes compatible with infant nutrition applications. The beneficial effects of combinations of available HMOs on the gut barrier, the identification of specific HMO(s) differentially exerting that effect, as well as its role in terms of protection against inflammation, remain underexplored. Here, by using an in vitro model of two intestinal epithelial cell lines, one of which produces mucus, we investigated the effects of 6 industrially available HMOs (2′FL, 3′SL, 6′SL, LNnT, LNT and DFL) individually and in various combinations, on epithelial barrier function. Herein, we revealed novel benefits of HMO combinations on intestinal epithelial integrity and protection against pro-inflammatory cytokine stimuli. The generated evidence suggested that 2′FL is the strongest contributor to the beneficial effect of the HMO mixes.

Human milk is composed of about 200 oligosaccharides with varying chemical structures [24]. The 6 HMOs tested in this study are major representatives of the 3 HMO chemical structures present in human milk [12]. We first investigated the effects of the 6 selected HMOs combined to mimic their relative concentrations in human milk [20]. In a non-inflammatory setting, this combination of HMOs dose-dependently strengthened the barrier to small solutes and ions, as assessed by the trans-epithelial resistance of the cell monolayers, although it did not modify the limited permeability to the 4000 Da FD4 macromolecule through the intact cell monolayer. The HMO6-mediated TEER increase before challenge was associated with enhanced cell resistance against the cytokine-mediated disruption of epithelial barrier integrity, as demonstrated by a dose-dependent limitation of TEER decline and FD4 translocation compared to challenged monolayers treated with vehicle. Of note, despite being the most abundant carbohydrate in human milk, lactose did not provide any significant effect, reflecting the uniqueness of HMOs’ impact on epithelial barrier.

Most dosages used in this study exceeded the HMO levels found in human milk (~10 g/L—~25 g/L [25,26]). Nonetheless, these results are in line with a previous study from the group of Wu et al. who showed that a HMO extract isolated from human milk dose-dependently reinforces the barrier to both ions and to macromolecules in an in vitro model of epithelial barrier dysfunction [17]. Notably, in that study, the barrier protective effect of the HMO extract was achieved at a comparable dose as that of HMO blends used in our current study. It is worth underscoring that, whilst the study by Wu and colleagues similarly demonstrated the protective effect of a crude extract of HMOs from pooled human milk samples [17], our study was unique in that we observed similar protective effects with a blend of 6 HMOs of lower complexity, with a well-characterized composition and available for industrial use.

Furthermore, our study was able to dissect the relative contribution of individual HMOs on the beneficial effects of this blend. To efficiently test a variety of HMO combinations, we utilized a screening method, which has minimum rigorous experimental repeats. This posts limitations and caveats on results and conclusions; nonetheless, it is an effective method to test as many combinations as possible in a given time. We were not able to identify a striking pattern of synergistic effects among the different individual HMOs or their chemical characteristics. Instead, all the groups of 3 HMO and 5 HMO combinations conferred a significant protection against the inflammatory challenge, independently of their composition. This observation is remarkable as it suggests that different HMOs may have redundant functions, conferring a certain protection to the immature epithelial barrier of the breast-fed infant despite the large variations in HMO composition observed in human milk [20,25].

Interestingly, we observed that among the HMOs studied, 2′FL seems to critically contribute to the protective effects of the blends, as 2′FL removal from HMO6 significantly reduced the efficacy of this blend to prevent inflammation-driven barrier leakiness, even when the HMO5 combination of the remaining HMOs was tested at the same total HMO concentration as HMO6. In addition, 2′FL-containing ternary blends were the only HMO3 group leading to permeability values not significantly different from the unchallenged monolayers. Finally, when tested in isolation, 2′FL also provided the most significant, dose-dependent protection. Of note, the 2′FL effects were unlikely due to production process contaminants, as different 2′FL batches, including batches produced by different technologies (fermentation and chemical synthesis), gave comparable results. As previously mentioned, 2′FL is, on average, the most concentrated HMO in human milk, but it is also among the ones experiencing the largest inter-subject variations, as its presence or absence depends on maternal genetics. Due to polymorphisms, the gene coding for an enzyme required for 2′FL synthesis (fucosyltransferase-2; FUT2) is inactive in part of the population (2–35% depending on the geographical region) [25]. 2′FL is absent in milk from FUT2-negative women. By contrast, 2′FL is usually the most abundant HMO in milk from FUT2-positive women, although, even in this population, there is a high inter-individual variability in the concentration of this HMO [25]. Notably, the risk of gastro-intestinal infections was shown to be lower in breast-fed infants from FUT2 positive women [27], whereas the risk of allergic disease was reduced with increasing levels of 2′FL in milk [28]. We speculate that our results may, at least partially, explain these observations, as well as the protective effect of 2′FL against intestinal inflammation shown in experimental models of necrotizing enterocolitis [29].

Investigating the potential mechanisms underlying the modulatory effect of HMOs on permeability was not within the scope of this study. However, our data on HMO-mediated increased trans-epithelial resistance in the unchallenged monolayers, and the subsequent protection against the inflammatory challenge, may be linked to HMO-related modulation of the synthesis and/or composition of glycosylated structures associated with the epithelial barrier permeability. In line with this, previous studies showed that 2′FL can promote glycocalyx development in Caco 2 cells [30], and that a complex blend of HMOs can upregulate Muc2 expression in human neonatal enteroids and in mucus secreting cell lines, associated with increased TEER [17]. Furthermore, direct anti-inflammatory effects of HMOs previously shown in Caco 2 cultures [31] may have dampened the impact of the cytokine challenge on the barrier integrity. HMOs, in particular 2′FL, were shown to prevent inflammation trough inhibiting TLR4 signaling [29] and CD14 expression [32]. Other mechanistic studies are required to better understand the specific modulation of the barrier by 2′FL and the unspecific protection brought by the combinations of the other HMOs.

A direct interaction of intact HMO molecules with the intestinal mucosa is likely to occur in vivo, as substantial amounts of HMOs survive the gastro-intestinal passage and can be retrieved in feces of breast-fed infants [15,16]. Additionally, HMOs are partially fermented into metabolites in the colon by the intestinal microbiota. Hence, it is equally possible that HMOs indirectly affect the epithelial barrier through products of fermentation by the microbiota, and further studies are warranted to address this question.

A major limitation of our study lies in its in vitro nature, particularly on the relevance of immortalized cell monolayers as a model for the infant intestinal epithelial barrier. However, it is not possible to conduct a similar study in clinical settings, and it is unclear whether other preclinical, including in vivo, models may have higher predictive value. Enteroids derived from intestinal stem cells are being increasingly used as models more closely mimicking the cell type complexity and, possibly, the physiology of the intestinal epithelium compared with classical cell line models. However, these models require access to intestinal tissue, which, in the case of infants, is particularly difficult to obtain. By combining enterocyte- and goblet-like cells in the same monolayer, we have attempted to be closer to a physiological epithelium than other in vitro, largely used models such as single Caco2 monolayers. Nonetheless, properly designed clinical studies would need to confirm our findings.

## 5. Conclusion

In summary, this research further substantiates the key role of HMOs on intestinal barrier homeostasis. Using an in vitro epithelial culture model of barrier dysfunction, we demonstrated that combinations and individual HMOs have differential capacity to modulate the epithelial barrier permeability, and that 2′FL seems to be a key contributor to the protection of the barrier integrity during an inflammatory insult. Interestingly, the blend of 6 HMOs with different chemical structures was furthermore able to increase the tightness of the epithelium in non-challenged conditions. Further clinical and mechanistic studies would be needed to confirm the relevance of HMOs in reinforcing the epithelial barrier and in conferring resistance against inflammation-related epithelial barrier dysfunction. This study nevertheless suggests a value of complex blends of HMOs available at an industrial scale in conferring host benefits closer to those provided by human milk when breastfeeding is not possible or only partial. Importantly, the results emphasize the mediating role of these key components on the beneficial effects of human milk and encourages the value of breastfeeding.

## Figures and Tables

**Figure 1 nutrients-12-03047-f001:**
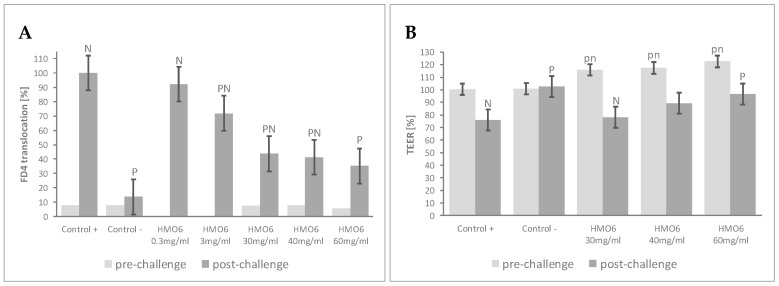
HMO6 confers resistance against inflammation-induced epithelial barrier dysfunction. Dose–response effect of HMO6 on (**A**) FD4 translocation (relative to Control + = 100%) at the end of the 48 h pre-challenge and post-challenge periods, and (**B**) TEER changes (relative to Time 0 h = 100%) during the 48 h pre-challenge and the 48 h post-challenge periods. Controls were non-HMO treated cells and cytokine unchallenged cells (Control −) and non-HMO treated cells, but cytokine challenged (Control +). Means ± 0.5 × LSD5% are presented, so that two conditions are significantly different if their error bars do not overlap. Significant differences vs. controls are highlighted (P = different from Control +, N = different from Control −), with capital letters for post-challenge and lowercase for pre-challenge. FD4: 4 KDa FITC-Dextran; HMO6: Blend of 6 human milk oligosaccharides; LSD_5%_: Fisher’s least significant difference with α = 5%.

**Figure 2 nutrients-12-03047-f002:**
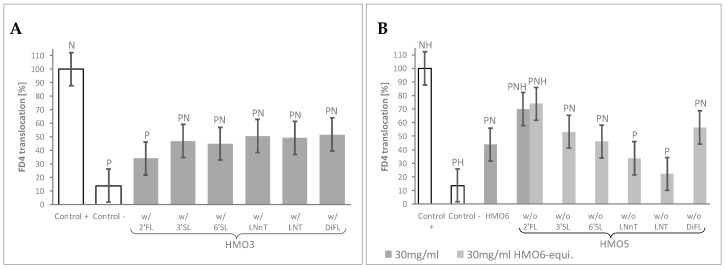
HMO sub-blends, particularly those containing 2′FL, protect the epithelial barrier. (**A**) Mean FD4 translocation of all ternary blends (HMO3) featuring the HMO reported in the horizontal axis, at 60 mg/mL HMO6-equivalent dose; (**B**) FD4 translocation with blends containing all except the HMO reported in the horizontal axis (HMO5), at 30 mg/mL HMO6-equivalent dose, and for HMO5 lacking 2′FL at 30 mg/mL as well. HMO6 data at 30 mg/mL are also represented. Controls were non-HMO treated cells and cytokine unchallenged cells (Control –) and non-HMO treated cells, but cytokine challenged (Control +). Means ± 0.5 × LSD5% are presented, so that two conditions are significantly different if their error bars do not overlap. Significant differences vs. controls are highlighted (P = different from Control +, N = different from Control −, H = different from HMO6). 2′FL: 2′Fucosyllactose; 3′SL: 3′Sialyllactose; 6′SL: 6′Sialyllactose; DFL: Difucosyllactose; FD4: 4 KDa FITC-Dextran; LNnT: Lacto-N-neotetraose; LNT: Lacto-N-tetraose; FD4: 4 KDa FITC-Dextran; HMO3 and HMO5: Blends of 3 and 5 human milk oligosaccharides, respectively; LSD_5%_: Fisher’s least significant difference with α = 5%.

**Figure 3 nutrients-12-03047-f003:**
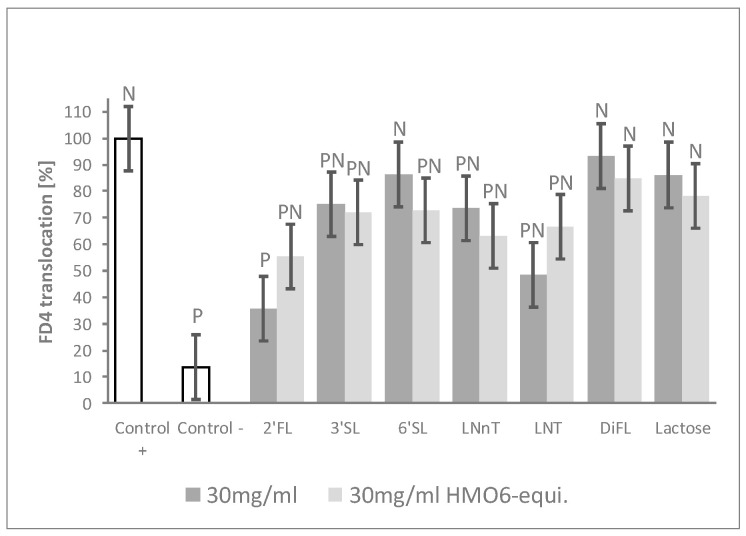
2′FL shows the highest barrier protection efficacy among individual HMOs. FD4 translocation with single HMO or lactose at both 30 mg/mL HMO6-equivalent dose and at 30 mg/mL. Controls were non-HMO treated cells and cytokine unchallenged cells (Control –) and non-HMO treated cells, but cytokine challenged (Control +). Means ± 0.5 × LSD5% are presented, so that two conditions are significantly different if their error bars do not overlap. Significant differences vs. controls are highlighted (P = different from Control +, N = different from Control −). 2′FL: 2′Fucosyllactose; 3′SL: 3′Sialyllactose; 6′SL: 6′Sialyllactose; DFL: Difucosyllactose; FD4: 4 KDa FITC-Dextran; LNnT: Lacto-N-neotetraose; LNT: Lacto-N-tetraose; LSD_5%_: Fisher’s least significant difference with α = 5%.

**Figure 4 nutrients-12-03047-f004:**
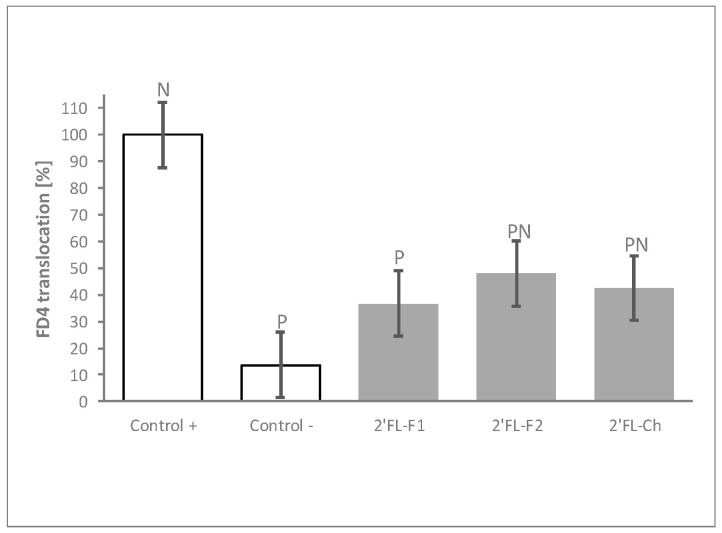
The process of 2′FL synthesis does not influence its protective effects. FD4 translocation with 2 different bacterial fermentation (2′FL-F1 and 2′FL-F2) and 1 chemical (2′FL-Ch) production batches of 2′FL at 30 mg/mL. Controls were non-HMO treated cells and cytokine unchallenged cells (Control –) and non-HMO treated cells, but cytokine challenged (Control +). Means ± 0.5 × LSD5% are presented, so that two conditions are significantly different if their error bars do not overlap. Significant differences vs. controls are highlighted (P = different from Control +, N = different from Control −). 2′FL: 2′Fucosyllactose; FD4: 4 KDa FITC-Dextran; LSD_5%_: Fisher’s least significant difference with α = 5%.

**Table 1 nutrients-12-03047-t001:** HMO concentration [mg/mL] and composition of the 49 treatments.^1^

Blocks	Treatments		2′FL	3′SL	6′SL	LNnT	LNT	DFL	Lactose
			[mg/mL]						
A	HMO6	0.3 mg/mL	0.165	0.021	0.027	0.015	0.054	0.018	0.000
		3 mg/mL	1.65	0.21	0.27	0.15	0.54	0.18	0.00
		30 mg/mL	16.5	2.1	2.7	1.5	5.4	1.8	0.0
		40 mg/mL	22.0	2.8	3.6	2.0	7.2	2.4	0.0
		60 mg/mL	33.0	4.2	5.4	3.0	10.8	3.6	0.0
B	HMO3_01	60 mg/mL HMO6-equivalent	33.0	4.2	5.4	0.0	0.0	0.0	0.0
	HMO3_02		33.0	4.2	0.0	3.0	0.0	0.0	0.0
	HMO3_03		33.0	4.2	0.0	0.0	10.8	0.0	0.0
	HMO3_04		33.0	4.2	0.0	0.0	0.0	3.6	0.0
	HMO3_05		33.0	0.0	5.4	3.0	0.0	0.0	0.0
	HMO3_06		33.0	0.0	5.4	0.0	10.8	0.0	0.0
	HMO3_07		33.0	0.0	5.4	0.0	0.0	3.6	0.0
	HMO3_08		33.0	0.0	0.0	3.0	10.8	0.0	0.0
	HMO3_09		33.0	0.0	0.0	3.0	0.0	3.6	0.0
	HMO3_10		33.0	0.0	0.0	0.0	10.8	3.6	0.0
	HMO3_11		0.0	4.2	5.4	3.0	0.0	0.0	0.0
	HMO3_12		0.0	4.2	5.4	0.0	10.8	0.0	0.0
	HMO3_13		0.0	4.2	5.4	0.0	0.0	3.6	0.0
	HMO3_14		0.0	4.2	0.0	3.0	10.8	0.0	0.0
	HMO3_15		0.0	4.2	0.0	3.0	0.0	3.6	0.0
	HMO3_16		0.0	4.2	0.0	0.0	10.8	3.6	0.0
	HMO3_17		0.0	0.0	5.4	3.0	10.8	0.0	0.0
	HMO3_18		0.0	0.0	5.4	3.0	0.0	3.6	0.0
	HMO3_19		0.0	0.0	5.4	0.0	10.8	3.6	0.0
	HMO3_20		0.0	0.0	0.0	3.0	10.8	3.6	0.0
C	HMO5 (w/o 2′FL)	30 mg/mL HMO6-equivalent	0.0	2.1	2.7	1.5	5.4	1.8	0.0
	HMO5 (w/o 3′SL)		16.5	0.0	2.7	1.5	5.4	1.8	0.0
	HMO5 (w/o 6′SL)		16.5	2.1	0.0	1.5	5.4	1.8	0.0
	HMO5 (w/o LNnT)		16.5	2.1	2.7	0.0	5.4	1.8	0.0
	HMO5 (w/o LNT)		16.5	2.1	2.7	1.5	0.0	1.8	0.0
	HMO5 (w/o DFL)		16.5	2.1	2.7	1.5	5.4	0.0	0.0
	HMO5 (w/o 2′FL)	30mg/mL	0.0	4.7	6.0	3.3	12.0	4.0	0.0
D	2′FL	30mg/mL HMO6-equivalent	16.5	0.0	0.0	0.0	0.0	0.0	0.0
	3′SL		0.0	2.1	0.0	0.0	0.0	0.0	0.0
	6′SL		0.0	0.0	2.7	0.0	0.0	0.0	0.0
	LNnT		0.0	0.0	0.0	1.5	0.0	0.0	0.0
	LNT		0.0	0.0	0.0	0.0	5.4	0.0	0.0
	DFL		0.0	0.0	0.0	0.0	0.0	1.8	0.0
	Lactose		0.0	0.0	0.0	0.0	0.0	0.0	16.5
	2′FL	30 mg/mL	30.0	0.0	0.0	0.0	0.0	0.0	0.0
	3′SL		0.0	30.0	0.0	0.0	0.0	0.0	0.0
	6′SL		0.0	0.0	30.0	0.0	0.0	0.0	0.0
	LNnT		0.0	0.0	0.0	30.0	0.0	0.0	0.0
	LNT		0.0	0.0	0.0	0.0	30.0	0.0	0.0
	DFL		0.0	0.0	0.0	0.0	0.0	30.0	0.0
	Lactose		0.0	0.0	0.0	0.0	0.0	0.0	30.0
	2′FL Ferment. 1	30 mg/mL	30.0	0.0	0.0	0.0	0.0	0.0	0.0
	2′FL Ferment. 2		30.0	0.0	0.0	0.0	0.0	0.0	0.0
	2′FL Chemical		30.0	0.0	0.0	0.0	0.0	0.0	0.0

^1^ 2′FL: 2′Fucosyllactose, 3′SL: 3′Sialyllactose, 6′SL: 6′Sialyllactose, DFL: Difucosyllactose HMO: Human milk oligosaccharides, HMO3, HMO5, HMO6: blends of 3, 5 and 6 HMOs, respectively, LNnT: Lacto-N-neotetraose, LNT: Lacto-N-tetraose.

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
