# Peer review of "Blends of Human Milk Oligosaccharides Confer Intestinal Epithelial Barrier Protection In Vitro"

_nutrients, 2020, doi:10.3390/nu12103047_

Round 1
Reviewer 1 Report
The authors used a Caco-2:HT29-MTX cell model to evaluate effects of individual and blends of human milk oligosaccharides on intestinal epithelial barrier protection. The manuscript is well written.
- Abstract, the cell model information needs to be included.
- Line 18, what are 3, 5, and 6 HMOs?
- Line 99, the information of the large intestine (caco-2:HT29-MTX=76:24) could not be found in reference 21.
- Which HMOs are bacterially produced and which HMOs are chemically-synthesized?
- Lines 133-138, how was FD4 measured?
- Line 149, how were the percentages of these six HMOs determined? Do these six HMOs appear similar concentrations in breast milk? Do the concentrations of different HMOs dynamically change during lactation?
- Base on Figure 1, the effects of HMOs on epithelial barrier are concentration dependent. However, for the 49 treatments (Table 1), the concentrations of HMOs blend or a single HMO are significantly different within the same block of experimental groups. For example, in block B, the concentrations of HMOs in HMO3_1 and HMO3_20 are remarkably different. Why lactose was not added to make the total concentration of HMOs the same for all the groups? How can results be interpreted from these experimental groups?
- The figure labels (Figure 1-4) for the statistical analyses are not clear.
- Since dietary effects were assessed in the current study, a high percentage of insults for the intestine are from the lumen side (apical side). Why the author did not treat the cells with reagents such as LPS and instead added TNF-alpha and IFN-gamma to the basolateral compartment to induce inflammation?
Minor:
- Line 91, is 10% Caco2 correct?
- Line 110, why a phenol red free medium was used for the treatments?
- Table 1 C block, is 30 mg/ml a typo?
- Line 20, FITC-labeled dextran: FD4
- The Abbreviations for controls without and with a cytokine challenge are hard to follow. What does ve mean? It would be better to show the control without a cytokine challenge first in the figure.
Author Response
The authors used a Caco-2:HT29-MTX cell model to evaluate effects of individual and blends of human milk oligosaccharides on intestinal epithelial barrier protection. The manuscript is well written.
1.Abstract, the cell model information needs to be included.
The cell model information is now included (line 23)
2.Line 18, what are 3, 5, and 6 HMOs?
This refers to the different HMO combinations described in table 1. Unfortunately, they cannot be listed here due to the limitation in abstract’s word number. However, we edited the text to be clear that varying combination of the 6 HMOs were tested (lines 19-21).
3.Line 99, the information of the large intestine (caco-2:HT29-MTX=76:24) could not be found in reference 21.
This is indeed a mistake. We verified that most publications with similar model suggest caco-2:HT29-MTX=70:30 as representative ratio found in large intestine (e.g. Ferraretto et al, Biosci Rep 2018). This is of course, not a ratio that we used. Instead we used a different ratio previously validated in our laboratory on transport and metabolism of food-derived bioactives such as ferulic acid. We have now corrected this statement in the manuscript and added the reference the publication we modeled the ratio used in our study (lines 101-102).
4.Which HMOs are bacterially produced and which HMOs are chemically-synthesized?
This point has been clarified in line 109.
5.Lines 133-138, how was FD4 measured?
Basolateral supernatants (100 μl in duplicate) were collected, and fluorescence was measured (Em: 485 nm; Ex: 535 nm; Tecan, Infinite 200). The FD4 concentration was calculated using a standard curve generated by serially diluting FD4 in culture media without phenol red. We have now included this additional information in the manuscript text (lines 138-143).
6.Line 149, how were the percentages of these six HMOs determined? Do these six HMOs appear similar concentrations in breast milk? Do the concentrations of different HMOs dynamically change during lactation?
The reviewer is right in that the concentrations of the HMOs used in this study vary along lactation. They are also very variable among lactating women. This variability in HMO contents and ratios is possibly biologically relevant but is difficult to reproduce in an experimental system. The ratios used in our study attempted to resemble the mean relative HMO concentrations in human milk, as stated in the discussion section, lines 313-314. This is now also mentioned in line 156.
7.Base on Figure 1, the effects of HMOs on epithelial barrier are concentration dependent. However, for the 49 treatments (Table 1), the concentrations of HMOs blend or a single HMO are significantly different within the same block of experimental groups. For example, in block B, the concentrations of HMOs in HMO3_1 and HMO3_20 are remarkably different. Why lactose was not added to make the total concentration of HMOs the same for all the groups? How can results be interpreted from these experimental groups?
We agree with the reviewer in that the total concentrations of HMOs differed among treatments in the HMO3 study and that this would have represented a major pitfall if the aim of this study would have been to compare the different HMO3 blends. However, as stated in lines 159-160, the aim of this screening assay was to estimate the individual contribution of each HMO (at its fixed concentration, same as the one of HMO6) when present in combination with other HMOs. These 20 blends represent all possible combinations of 3 HMOs out of 6. This allows aggregating them to provide unbiased estimates of the desired individual contributions, which would not have been the case if using lactose (or any other compound) as a filler.
8.The figure labels (Figure 1-4) for the statistical analyses are not clear.
The statistical part of the figure labels has been simplified and aligned with section “2.4. Experimental design & data analyses”.
9.Since dietary effects were assessed in the current study, a high percentage of insults for the intestine are from the lumen side (apical side). Why the author did not treat the cells with reagents such as LPS and instead added TNF-alpha and IFN-gamma to the basolateral compartment to induce inflammation?
The aim of our study was to investigate the protective effect of HMOs against inflammation-related gut barrier-disruption, which occurs upon release of pro-inflammatory cytokines by immune cells in the lamina propria. The TNF-alpha and INF-gamma challenge is a robust, widely used model that in our opinion enables addressing this aim. We agree in that apical LPS could have been a good model to mimic the initial insult by gram negative bacteria and we similarly tested this model during our preliminary experiments. However, we were unable to observe any effect of apical LPS stimulation even using a high concentration of 100 µg/ml on monolayer permeability. The unresponsive of our cell lines may not be unique to our model, indeed similar reports have been observed by other groups, in particular, specific Caco2 cell lines unresponsiveness to apical LPS (e.g. Funda et al. INFECTION AND IMMUNITY 2001, 69 (6): 3772–3781; Schuerer-Maly et al. Immunology 1994, 81: 85-91).
Minor:
1.Line 91, is 10% Caco2 correct?
10% CO2, 95% air/water saturated atmosphere were indeed the environmental conditions used in the experiments
2.Line 110, why a phenol red free medium was used for the treatments?
Phenol red free medium was used during the treatment periods to prevent any interference with the FD4 analysis.
3.Table 1 C block, is 30 mg/ml a typo?
This is right (not a typo). As reported in Figure 2B and stated in lines 251-254, the sub-blend HMO5 without 2FL was tested in 2 conditions, at 30 mg/ml HMO6 equivalent dose and at 30 mg/ml total HMO concentration.
In addition, please note that the labels of the HMO5 blends in table 1 and in Supplementary table have been changed to align with those used in Figure 2B
4.Line 20, FITC-labeled dextran: FD4
Thank you for noticing this typo. It has been changed accordingly
5.The Abbreviations for controls without and with a cytokine challenge are hard to follow. What does ve mean? It would be better to show the control without a cytokine challenge first in the figure.
“ve” has been eliminated from the labels. Positive control is now labelled as “Control +” and negative control as “Control –“.
Reviewer 2 Report
This is a review of the manuscript entitled "Blends of Human Milk Oligosaccharides Confer Intestinal Epithelial Barrier Protection in vitro." In this study, the authors aim to determine the influence various HMOs have on barrier function. Six HMOs were chosen as they were highly represented in human milk and were industrially available for study. The authors first established an optimal dose to test and then examined the effect of these after cytokine challenge. They then found 2'FL appeared to be primarily responsible for protection against cytokine challenge and that it mediated its effect irrespective of its source. In general, the manuscript is well written, the research question is clearly defined and carried out appropriately and its limitations are stated. I would recommend for publication and have only slight points to raise.
- The statistics need more clarification is this. Were changes in permeability in Fig 1-4 significant or not?
- Although it is stated in the methods, it would do no harm to repeat in the results section that there was a cell death assay carried out and less than 5% of cells died during cytokine challenge
- There should be a brief commentary in the discussion about the use of human organoids/monolayers derived from human intestinal organoids for these types of studies.
Author Response
This is a review of the manuscript entitled "Blends of Human Milk Oligosaccharides Confer Intestinal Epithelial Barrier Protection in vitro." In this study, the authors aim to determine the influence various HMOs have on barrier function. Six HMOs were chosen as they were highly represented in human milk and were industrially available for study. The authors first established an optimal dose to test and then examined the effect of these after cytokine challenge. They then found 2'FL appeared to be primarily responsible for protection against cytokine challenge and that it mediated its effect irrespective of its source. In general, the manuscript is well written, the research question is clearly defined and carried out appropriately and its limitations are stated. I would recommend for publication and have only slight points to raise.
1.The statistics need more clarification is this. Were changes in permeability in Fig 1-4 significant or not?
As described in section “2.4. Experimental design & data analyses” and shortly indicated in the legends of figures 1-4, two conditions are significantly different if their error bars do not overlap. This alternative way of presenting post-hoc comparisons allows to simplify the figures while providing the same information as would be given by extra symbols (Mason, Gunst, Hess, 2003). The same type of figures is presented by e.g. Sabatier et. al. (2020) Impact of Ascorbic Acid on the In Vitro Iron Bioavailability of a Casein-Based Iron Fortificant, Nutrients, Volume 12, Issue 9.
2.Although it is stated in the methods, it would do no harm to repeat in the results section that there was a cell death assay carried out and less than 5% of cells died during cytokine challenge
A sentence has been added to the result section accordingly (lines 198-199).
3.There should be a brief commentary in the discussion about the use of human organoids/monolayers derived from human intestinal organoids for these types of studies.
We thank you the reviewer for the suggestion. This topic is now discussed in lines 387-391.
Round 2
Reviewer 1 Report
All the questions have been addressed.